# *LmCht5-1* and *LmCht5-2* Promote the Degradation of Serosal and Pro-Nymphal Cuticles during Locust Embryonic Development

**DOI:** 10.3390/biology11121778

**Published:** 2022-12-07

**Authors:** Tingting Zhang, Yanjun Huo, Qing Dong, Weiwei Liu, Lu Gao, Jiannan Zhou, Daqi Li, Xueyao Zhang, Jianzhen Zhang, Min Zhang

**Affiliations:** 1Research Institute of Applied Biology, Shanxi University, Taiyuan 030006, China; 2Institute of Plant Protection, Shanxi Academy of Agricultural Science, Taiyuan 030031, China

**Keywords:** *Locust migratoria*, serosal cuticle, pro-nymphal cuticle, Chitinase 5-1, Chitinase 5-2

## Abstract

**Simple Summary:**

This paper focus on the role of chitinase 5-1 (*LmCht5-1*) and chitinase 5-2 (*LmCht5-2*) in the degradation of the serosal cuticle and pro-nymphal cuticle during locust embryonic development. Serosal cuticles degenerated from 7-day-old embryos (E7) to E13 with the degradation of fine chitin in the lower part and coarse chitin layer in upper part in sequence, while pro-nymphal cuticles degenerated from E12 to E14 with degradation of chitin in the procuticle layer. RNAi experiments in embryonic stages showed that during the serosal cuticle molting process, ds*LmCht5-1* averted the loss of the coarse chitin layer in the upper part during early and late embryogenesis. Meanwhile, ds*LmCht5-2* blocked the degradation of the lower fine chitin layer at the early stage and blocked the chitin degradation of loose coarse chitin in the late molting process. During pro-nymphal cuticle degradation stages, ds*LmCht5-1* inhibited the degradation of chitin between layers, and ds*LmCht5-2* inhibited the degradation of chitin into filaments within layers. This study advances the understanding of the degradation of cuticles in the locust embryonic stage and makes a contribution to molecular targets for agricultural pest control.

**Abstract:**

The success of the degradation of the extraembryonic serosal cuticle and the second embryonic cuticle (pro-nymphal cuticle) is essential for the development and molting of nymph from egg in Orthoptera *Locusta migratoria*. Chitinase 5 is an important gene for chitin degradation in nymphs and in the egg stage. In this study, we investigated the important roles of chitinase 5-1 (*LmCht5-1*) and chitinase 5-2 (*LmCht5-2*) in the degradation of the serosal and pro-nymphal cuticles during locust embryonic development. The serosal cuticle degrades from 7-day-old embryos (E7) to E13, along with the degradation of the pro-nymphal cuticle, which begins at E12 to E14. The mRNA and protein of *LmCht5-1* and *LmCht5-2* are expressed during the degradation process of the serosal cuticle and the pro-nymphal cuticle. RNAi experiments at the embryonic stage show that both ds*LmCht5-1* and ds*LmCht5-2* contribute to the failure of development in early and late embryogenesis. Further, during the serosal cuticle molting process, ultra-structure analysis indicated that ds*LmCht5-1* prevented the loss of the coarse chitin layer in the upper part in both early and late embryogenesis. Meanwhile, ds*LmCht5-2* blocked the degradation of the lower fine chitin layer at the early stage and blocked the chitin degradation of loose coarse chitin in the late molting process. During the degradation of the pro-nymphal cuticle, ds*LmCht5-1* suppresses chitin degradation between layers in the procuticle, while ds*LmCht5-2* suppresses chitin degradation into filaments inside of the layer. In summary, our results suggest that both *LmCht5-1* and *LmCht5-2* contribute to the degradation of the serosal and pro-nymphal cuticles during the locust embryonic stage.

## 1. Introduction

Molting, which occurs not only during the postembryonic period but also the embryogenesis of insects [1], is often considered to be an important biological process in pest control [2]. As a representative of hemimetabolous insects, the worldwide agricultural pest *L. migratoria* develops from an egg to the adult stage, going through five nymphal stages and no pupal stage. It sheds the old cuticle and promotes a new cuticle at the end of each nymph phase. Moreover, there are four cuticle formation times and three cuticle degradation times during egg development, which match with four peaks of molting hormone 20-hydroxyecdysone (20E) both in *Locusta migratoria* and *Schistocerca gregaria* [3,4,5]. The extraembryonic serosal cuticle is the first cuticle produced in three-day-old embryos (E3) and begins to degrade at E7 [6]. Next, the first embryonic cuticle covers the embryo at E8 to E9 without the typical chitin structure. Then, the embryo secretes the second embryonic cuticle. Since it was deposited before the “nymphal” cuticle, Truman and Riddiford also named it the “pro-nymph” cuticle [7]. The pro-nymphal cuticle is secreted at E9 and degrades from E11 to hatching at E13-E14 [3]. At stage E13, the nymph will be covered by the fourth embryonic cuticle, or the first nymphal cuticle [8]. Therefore, the degradation of the serosal cuticle and the pro-nymphal cuticle are the typical hydrolysis of the cuticle during the embryogenesis of the locust, which could be a potential stage for pest control.

Generally, the insect cuticle, including the nymph cuticle and the embryonic cuticle, is the extracellular structure that supports and protects the embryo [9]. They support the muscle attachments needed for locomotion as the exoskeleton, prevent physical and chemical damage, and protect against infection and water loss as a barrier [10,11,12,13]. The main component of the insect cuticle, including the serosal [14] and the pro-nymphal cuticle [2], is chitin, which is a β-(1,4)-linked configuration of products [15] thatis hydrolyzed to low molecular oligosaccharides before each molting.

Insect chitinases belong to the glycoside hydrolase 18 family [16], among which the type I chitinase gene (Cht5s) is mainly involved in the digestion of the old cuticle during insect molting. As an endonuclease, it can degrade chitin to chitin-oligosaccharide. The typical structure contains a signal peptide region, a catalytic domain, a chitin binding domain, and a serine/threonine rich glycosylated link region [17]. RNAi of *Tribolium castaneum* Cht5 (*TcCht5*) has shown that this gene is critical for insect molting and growth [18,19]. Similarly, *SeCht5* in *Spodoptera exigua* molting was required for pupation and emergence [20]. In our previous study, we found two chitinase 5 genes in *L. migratoria,* which we named *LmCht5-1* and *LmCht5-2*. However, *LmCht5-1* has a chitin binding domain (CBD), while *LmCht5-2* does not have one [2,21]. RNAi-mediated suppression of the *LmCht5-1* transcript led to severe molting defects and lethality from the fifth instar nymph to adult, but no observed phenotypes were found in the *LmCht5-2* RNAi group in *L. migratoria* [22]. The ds*LmCht5-1* was injected by microinjection during the development of locust embryo E9, which caused the block or failure of procuticle degradation and nymph hatching [3], while ds*LmCht5-2* injected at E5 and E7 inhibited the degradation of the serosal cuticle in our unpublished data, suggesting *LmCht5-1* and *LmCht5-2* play diverse molecular roles in embryonic cuticle degradation and could be the molecular target for pest control in the egg stage.

Although the function of Cht5 genes in embryonic cuticle degradation has been reported, the comprehensive function and differentiation of *LmCht5-1* and *LmCht5-2* on the serosal cuticle and the pro-nymphal cuticle has not been researched. Specifically, we analyzed the mRNA expression patterns of *LmCht5-1* and *LmCht5-2* during the egg stage. Dynamic expression and localization of *LmCht5-1* and *LmCht5-2* were observed using modified paraffin profiles and immunohistochemical staining. The function and differentiation of *LmCht5-1* and *LmCht5-2* on the degradation of chitin in the serosal cuticle and the pro-nymphal cuticle were analyzed by transmission electron microscopy (TEM) using embryonic RNAi technology.

## 2. Materials and Methods

### 2.1. Insects

*L. migratoria* was cultured in a single well-ventilated climate chamber in an insect incubator room at 30 ± 1 °C and 50% relative humidity (RH) at the Institute of Applied Biology, Shanxi University. Adults were reared in a cage (30 cm × 30 cm × 30 cm) with a 16:8 light to dark photo-cycle and fed with wheat seedlings and wheat bran. Eggs were collected daily in disposable petri dishes and placed in boxes sealed with a preservative film and perforated to ensure air circulation during incubation. Eggs were incubated at 30 ± 1 °C and 50% RH. Eggs after day 5 of development (E5) and E7 were followed up by microinjection to study their hatching rate.

### 2.2. LmCht5 Expression Pattern during Embryogenesis

To investigate expression patterns of *LmCht5-1* and *LmCht5-2* in embryonic development, we collected egg samples from 8-hour-old eggs (8H) to 14-day-old eggs (E14) and collected every 12 h. Five biological replicates were prepared for each stage. Total RNA was extracted using Trizol Plus (TaKaRa, Tokyo, Japan) according to the manufacturer’s instruction. The total RNA product from each biological copy was evaluated and quantified on 1% agarose gel. The first-strand cDNA was synthesized using a cDNA synthesis kit (Vazyme, Nanjing, China). Each cDNA sample was diluted five times for RT-qPCR analysis. RT-qPCR analyses for the expression level of *LmCht5-1* and *LmCht5-2* were performed using an ABI 7300 Real-Time PCR System (Applied Biosystems Inc., Foster City, CA, USA). The gene-specific and *LmRpl32* primers were designed for RT-qPCR, and their sequences are shown in Table 1. *LmRpl32* is used as an internal gene. Five independent biological and two technical repetitions were performed for each sample.

### 2.3. dsRNA Synthesis and Microinjection

To investigate the function of *LmCht5-1* and *LmCht5-2* in embryogenesis, dsRNA targeted gene-specific primers were designed according to the E-RNAi online website (http://www.dkfz.de/signalling/e-rnai3/, accessed on 16 September 2016) (Table 1) and synthesized by Thermo Fisher Scientific (China, Shanghai). In brief, dsRNA against *LmCht5-1*, *LmCht5-2* and *dsGFP* was synthesized as previously described [3]. The final concentration of dsRNA was adjusted to 2 μg/μL for microinjection. The liquid of ds*GFP*, ds*LmCht5-1*, and ds*LmCht5-2* was injected according to the method described by Zhang et al. [3]. The RNAi experiment was repeated three times in each group with 60 eggs injected each time. RNAi efficiency was measured at 24 h after injection. The remaining eggs continued to be cultured under the same conditions.

### 2.4. Paraffin Sections of Locust Eggs

The paraffin section of locust eggs was prepared as described before [2]. For analysis of the function of *LmCht5-1* and *LmCht5-2* at the early degradation stage, eggs were collected at E7 after 48 h of dsRNA injection. For analysis of the function of *LmCht5-1* and *LmCht5-2* at the late degradation stage of the serosal cuticle and the pro-nymphal cuticle stage, eggs were collected at E13 and the dsRNA was injected at E7. Five eggs were collected from each replication.

### 2.5. Immunofluorescence Microscopy

To investigate the localization of *LmCht5-1* and *LmCht5-2* during the degradation of the serosal cuticle and the pro-nymphal cuticle, immunohistochemistry was performed as follows. Sections were deparaffinized with xylene twice, 10 min each time; and rehydrated in an ethanol gradient at 5 min each step (100%, 95%, 90%, 80%, 70%, 50%, 30% and double-distilled water). Samples were treated with 3% H_2_O_2_ to inactivate endogenous peroxidase, and then they were boiled in citrate buffer for 20-30 min for antigen recovery. After blocking with 3% BSA for 30 min, sections were probed with *LmCht5-1* or *LmCht5-2* specific antibodies (1:100) separately [23]. After washing with PBS, the secondary antibody (1:200) with fluorophorecoupling was incubated for 2 h at 37 °C. Subsequently, sections were incubated with 1 mg/mL Fluorescent Brightener 28 (FB28, Sigma-Aldrich, St. Louis, MO, USA) for 30 s to label chitin [3]. After washing three times with PBS, finally, the nucleus was stained with SYTOX^®^Green nucleic for 15 min. Staining was viewed using a magnification of 60X on an LSM 880 confocal laser scanning microscope (Zeiss, Germany).

### 2.6. Transmission Electron Microscopy

Transmission electron microscopy (TEM) technology was used to further observe the ultrastructure of the serosal cuticle and the pro-nymphal cuticle. The specimens for dsRNA function on degradation of the serosal cuticle were collected on E8 and E13. Meanwhile, the specimens for dsRNA function on pro-nymphal cuticle degradation were collected on E13. These samples were fixed in 3% glutaraldehyde. Ultrathin section preparation and imaging were performed according to the method described by Yu et al. [24]. Ultrathin section images of treated ds*LmCht5-1*, ds*LmCht5-2*, or ds*GFP* embryos were captured using a JEM-1200EX transmission electron microscope (TEM, JEOL, Tokyo, Japan).

### 2.7. Statistical Analysis

Differences between developmental stages were analyzed by one-way ANOVA by Tukey’s HSD multiple comparison test (IBM SPSS Statistics 18.0, IBM Corp., Armonk, NY, USA). The two sets of data were statistically analyzed by independent sample Student’s *t* test. Asterisks indicate significant differences (* *p* < 0.05; ** *p* < 0.01).

## 3. Results

### 3.1. Ultra-Structure Showed Significant Differences in Cuticle Degradation between the Serosal Cuticle and the Pro-Nymphal Cuticle

The degradation of the extraembryonic serosal cuticle and the intraembryonic pro-nymphal cuticle is essential for the embryonic development and hatching of the locust. We first used TEM technology to observe the difference of ultrastructure between the serosal cuticle and the pro-nymphal cuticle during degradation (Figure 1). The serosal cuticle is composed of two layers of patchy chitin with uneven electron density, of which the upper part showed an atypical coarse chitin arrangement layer with large block structure and the lower part showed an atypical fine, sparsely dispersed, filamentous, electron-dense material at the E6 stage (Figure 1A_1_). However, the intra-embryonic typical pro-nymphal cuticle is composed of a single layer epicuticle on the upper part and a multilayer, densely packed chitin layer procuticle on the lower part at E11 (Figure 1B_1_). The degradation of the serosal cuticle began at E7 and was completed at E14 (Figure 1A), while the degradation process of the pro-nymphal cuticle was from E12 to E14 (Figure 1B). When the degradation process of the serosal cuticle began, the fine, sparsely filamentous chitin on the lower part hydrolyzed to disappear, and the thick chitin layer arrangement with large block structure become loose at E7 (Figure 1A_1_,A_2_). Then the thick chitin layer hydrolyzed to smaller chitin electronic material with tighter, thinner electron density (Figure 1A_3_,A_4_) and finally disappeared until E14 (Figure 1A_5_). The degradation process of the pro-nymphal cuticle started with the apolysis between the epidermis cell and procuticle and the separation of the epicuticle and procuticle at E12 (Figure 1B_1_,B_2_). Then the multilayer, densely packed chitin hydrolyzed to small pieces of filamentous chitin layer by layer (Figure 1B_3_,B_4_) until only the epicuticle was left at the E14 stage (Figure 1B_5_).

### 3.2. LmCht5-1 and LmCht5-2 Development Expression Patterns during Embryogenesis

To investigate the roles of *LmCht5-1* and *LmCht5-2* during the cuticle degradation process, we used the RT-qPCR method to determine their developmental expression patterns during embryonic development. The transcription level of *LmCht5-1* was under the low level in the early stage of embryonic development (8H-E9H0), reaches its peak from E9H12, and then maintains a high expression level after E11H12 (Figure 2A). The expression level of the *LmCht5-2* transcript was low in early embryonic development (8H-E5H12), gradually increased after E6H0, reached a peak at E8H0, and then decreased and maintained a low expression level. The degradation process of the serosal cuticle occurred with the high expression level of *LmCht5-2* first and later the high expression level of *LmCht5-1*, while the degradation of the pro-nymphal cuticle was consistent with the high expression level of *LmCht5-1* and low expression level of *LmCht5-2* (Figure 2A,B).

### 3.3. Both dsLmCht5-1 and dsLmCht5-2 Cause Lethality during Early and Late Embryogenesis of the Locust

In order to analyze whether *LmCht5-1* and *LmCht5-2* are important in the serosal cuticle degradation stage, we microinjected *LmCht5-1* and *LmCht5-2* specific dsRNA into eggs at E5. The RT-qPCR detected after 24 h of each specific target dsRNA microinjection showed that the expression of *LmCht5-1* and *LmCht5-2* was reduced 77.0% and 89.5% at the mRNA level compared with the ds*GFP* group, respectively (Appendix A). Almost all of the embryos in the ds*GFP* group could successfully grow to the katatrepsis stage, which includes typical development characteristics in early embryogenesis. However, 48.8% of embryos in the development ds*LmCht5-1* group (Appendix A) and 61.7% of embryos in the ds*LmCht5-2* group (Appendix A) were delayed or blocked at the katatrepsis stage, indicating that the interference of *LmCht5-1* and *LmCht5-2* in the early stage of serosal cuticle degradation could hinder the normal development of locust embryos. A similar microinjection of specific target dsRNA on Cht5s was performed at E7 to evaluate its effect on the degradation of the pro-nymphal cuticle in the hatching stage during late embryogenesis. The RT-qPCR detected after 24 h revealed that the expression of *LmCht5-1* and *LmCht5-2* was downregulated by 82.3% and 72.5% at the mRNA level compared with the ds*GFP* group, respectively (Appendix A). Further, 83.3% of embryos in the ds*LmCht5-1* group (Appendix A) and 78.3% of embryos in the ds*LmCht5-2* group (Appendix A) were delayed or blocked at the hatching stage (Appendix A), suggesting that inhibition of *LmCht5-1* and *LmCht5-2* in the degradation of the pro-nymphal cuticle could also reduce the hatch rate of normal locust eggs.

### 3.4. LmCht5-1 and LmCht5-2 Proteins Were Expressed during Serosal Cuticle and Pro-Nymphal Cuticle Degradation

To further understand the potential function of Cht5s in the degradation of the embryonic cuticle, we first used *LmCht5-1* and *LmCht5-2* polyclonal antibodies to perform immunohistochemical staining on paraffin sections to locate their expression. During the serosal cuticle degradation process, the *LmCht5-1* protein was expressed at a low level at E7, its expression highly scattered in the *LmCht5-1*. Meanwhile, *LmCht5-2* was mainly expressed in the fine structure of the lower serosal cuticle from E7 to E8 and later in the looser, coarser cuticle at E9, and finally expressed at a low level in the serosal cuticle until hatching (Figure 3A). However, in the metabolism of the pro-nymphal cuticle, *LmCht5-1* was first expressed in the epidermis cell at E9 then secreted into the procuticle with the increased level at E11, and still expressed in the old pro-nymphal cuticle with a gradually decreasing concentration. At the same time, *LmCht5-2* was always expressed at a low level in the pro-nymphal cuticle and had a considerable expression level at E11, which is the point for starting of degradation of the pro-nymphal cuticle (Figure 3B). The location and expression of *LmCht5-1* and *LmCht5-2* suggested that they may play essential roles in degradation of both the serosal cuticle and the pro-nymphal cuticle.

### 3.5. dsLmCht5-1 and dsLmCht5-2 Inhibited Cuticle Degradation of the Serosal Cuticle in Both Early and Late Embryogenesis of the Locust

We then sought to understand the katatrepsis or molting blocking phenotype caused by RNAi of *LmCht5-1* and *LmCht5-2*, respectively. Paraffin sections of eggs at stage E8 or E13 of ds*LmCht5-1*, ds*LmCht5-2*, and ds*GFP* injected groups were prepared for an immunohistochemistry test (Figure 4A–D and Figure 5A,B). At stage E8, the fine chitin structure on the lower part of the serosal cuticle has been completely hydrolyzed, and the coarse chitin structure on the upper part is in a relaxed state, with comparable expression levels of both *LmCht5-1* and *LmCht5-2*. Compared to the ds*GFP* group, the ds*LmCht5-1* injection group showed a reduced expression level of *LmCht5-1* and a much tighter layer in the coarser chitin portion. Meanwhile, the ds*LmCht5-2* injection group displayed the fine chitin structure hydrolyzation delay or block accompanied by the decreasing expression of *LmCht5-2* on both fine and coarser chitin structures (Figure 4A,B). Further, the TEM analysis of the serosal cuticle at E8 showed that the electron density in the upper layer in the ds*LmCht5-1* group was denser than that in the control group, while the fine chitin structure in the lower layer in the ds*LmCht5-2* group was not degraded (Figure 5A). At stage E13, the serosal cuticle has been almost completely hydrolyzed, leaving a small, loose chitin layer and barely visible expression levels of *LmCht5-1* and *LmCht5-2*. The degradation of the coarser structure chitin in the upper part of the serosal cuticle was not obviously affected by ds*LmCht5-1* and ds*LmCht5-2* in the immunohistochemistry analysis (Figure 4C,D). However, in the late embryonic developmental by TEM analysis showed that ds*LmCht5-1* still suppresses the loss of coarser structures from the upper layer of the serosal cuticle, while ds*LmCht5-2* inhibits the further degradation of looser, coarser structures at E13 (Figure 5B). Therefore, both *LmCht5-1* and *LmCht5-2* are required for serosal cuticle degradation during katatrepsis in early embryogenesis and molting in late embryogenesis.

### 3.6. dsLmCht5-1 and dsLmCht5-2 Inhibited Chitin Degradation of the Pro-Nymphal Cuticle

To further confirm the role of Cht5s in the degradation of the pro-nymphal cuticle, we analyzed immunization and TEM tests at stage E13. At stage E13, the ds*LmCht5-1* and ds*LmCht5-2* injection groups showed reduced levels of protein expression and appeared to be normal when analyzed. However, the pro-nymphal cuticle in both the ds*LmCht5-1* and ds*LmCht5-2* groups showed a thicker cuticle to a certain extent than that of the control group (Figure 4E,F). Moreover, the TEM result indicated that the procuticle in the ds*LmCht5-1* group was in the regular compact layer, compared with only some chitin filaments in the procuticle in the control group (Figure 5C). Meanwhile, the procuticle in the ds*LmCht5-2* group had a regular layer cuticle with loose visible chitin filaments, indicating that ds*LmCht5-2* also inhibits the degradation of the chitin structure inside of the layer in the procuticle but to a lesser extent than that of ds*LmCht5-1* (Figure 5C). As a result, both *LmCht5-1* and *LmCht5-2* are involved in the product degradation of the procuticle cuticle during the molting process in late embryogenesis.

## 4. Discussion

Many insects shed their cuticle during the embryonic stage, including Zygentoma, Ephemeroptera, Neuroptera, Phasmatodea, Mantodea, Blattodea, and Orthoptera [25]. *L. migratoria* as representatives of Orthoptera insects have large egg bodies and obvious and different types of embryonic cuticles, which can be used as a crucial model organism to study the occurrence of embryonic cuticles, and are ideal materials to study the mechanism of chitin metabolism in the inner cuticle of eggs [26].

The serosal cuticle and the pro-nymphal cuticle are considered to be the visible chitin cuticle in early and late embryogenesis of the locust, respectively. To obtain a greater understanding of the functions of chitinases on embryonic cuticles, we analyzed the function of Cht5, the first group I chitinases that had been reported in a variety of insect species [16,27,28,29,30]. Specifically, the two Cht5s in the locust have different functions on nymph-to-nymph molting. Down-regulation of *LmCht5-1* causes death before nymph molting with failure of cuticle degradation, while reduced the expression of *LmCht5-2* does not cause any visible phenotype in the nymph. However, down-regulation of *LmCht5-1* and *LmCht5-2* in the early and late stages of embryogenesis also leads to the blocking and delay of embryonic development, like katatrepsis and molting. Likewise, in *S. exigua*, it is shown that *SeCht5* is responsible for chitin degradation during pupation and eclosion [20]. Consistently, we also found that *LmCht5-1* was essential for the degradation of the procuticle in the pro-nymphal cuticle but not for apolysis [3]. Taken together, these results suggest that Cht5s has the relative conserved function as the fundamental role in the chitin degradation.

Using TEM analysis and the reported structure in HE analysis of the serosal cuticle [28], we conclude that the normal degradation of the serosal cuticle is extremely critical for insect embryo development. During early embryogenesis, first, the fine structure chitin in the low part of the serosal cuticle is hydrolyzed, and the coarse chitin in the layer on the upper part becomes loose from bottom to top; second, the loose, coarse chitin in the bottom hydrolyzes into chitin filaments. Third, the rest of coarse chitin in the top continues to loosen and hydrolyze into chitin filaments in late embryogenesis (Figure 6) [6]. The cuticle of ds*LmCht5-1* injected embryos was examined by a series of immunostaining revealed that *LmCht5-1* is required for coarse chitin loss in the upper part of the serosal cuticle in both early and late embryogenesis (Figure 5A,B) but little function on the fine structure degradation in the low part (Figure 5A). Meanwhile, *LmCht5-2* is mainly involved in the degradation of the fine chitin structure degradation in early embryogenesis (Figure 5A) but has little or no function on the loss of coarse chitin (Figure 5B), which is also confirmed using HE and immunohistochemical technology [6].

For the degradation of the pro-nymphal cuticle during molting, we confirmed the three previously-known steps: separation of the epicuticle from the procuticle, apolysis from the epidermis, and then procuticle degradation [3]. *LmCht5-1* has been reportedly involved in the degradation of the procuticle but not for apolysis [3]. Using TEM analysis every 12 h in late embryogenesis, the procuticle degradation could be further divide into the loss of chitin layers and the degradation of chitin into filaments inside the layer (Figure 6). The immunofluorescence and TEM experiments on the pro-nymphal cuticle after ds*LmCht5-1* or ds*LmCht5-2* injection indicated that *LmCht5-1* was mainly responsible for the chitin layer loss but bore little or no responsibility for chitin degradation inside the layer. However, *LmCht5-2* mainly promoted chitin degradation into filaments inside the layer but not degradation between the chitin layers (Figure 5C).

Combining the degradation of the serosal cuticle and the pro-nymphal cuticle during embryogenesis, we can propose that the degradation process of the cuticle is first the loosening of the connection between layers and then the hydrolysis of layers chitin into filaments inside the layer in which either the cuticle is typical or has an atypical layer structure. Using the specific antibodies of *LmCht5-1* and *LmCht5-2*, we found that they both were expressed in all embryonic cuticles but with different levels. *LmCht5-1* with the CBD [22] is mainly responsible for the loss of the chitin connection between layers, either in coarse chitin of the serosal cuticle or in the typical layer chitin of the pro-nymphal cuticle. Meanwhile, *LmCht5-2* does not have a CBD for binding to colloidal chitin, which mainly promotes chitin degradation into filaments inside layers, such as for the fine structure degradation in the serosal cuticle and chitin degradation inside the procuticle layer. Consistently, in vitro expression and the chitin degradation test of *LmCht5-1* and *LmCht5-2* showed that both of them could degrade oligo-chitin substrate 4MU-(GlcNAc)_3_ and poly-chitin substrate CM-chitin-RBV simultaneously [31]. Likewise, the vitro expression of *OfCht5* in *Ostrinia furnacalis* indicated that *OfCht5* could hydrolyze the colloidal chitin, α-chitin, and (GlcNAc)_6_ [32,33,34]. However, *LmCht5-1* tended to degrade the oligo-chitin substrate 4MU-(GlcNAc)_3_, and *LmCht5-2* is unable to bind gelatinous chitin and tends to degrade CM-chitin-RBV substrates [31]. Hence, additional experiments for identification of chitin types inside or between layers is needed for a comprehensive understanding of the chitin degradation and function of chitinases.

Therefore, our results showed that *LmCht5-1* and *LmCht5-2* are involved both in the serosal cuticle degradation during E7 to E14 and the pro-nymphal cuticle degradation during E11 to E14. The katatrepsis that occurred at E6-E7 is the typical embryonic development stage, accompanying the dramatic turnover of the embryo and the extramembrane change. The process was blocked or delayed by the inhibition expression of *LmCht5-1* and *LmCht5-2*, which suggested that the degradation of the serosal cuticle inhibits katatrepsis or even leads to death of the locust. Another important development stage is molting, which happened during E13 and E14, accompanied by the degradation of little of the serosal cuticle and most of the pro-nymphal cuticle in both the locust and *Nilaparvata lugens* [3,35]. In this process, we found *LmCht5-1* and *LmCht5-2* promoted the degradation of both the serosal cuticle and the pro-nymphal cuticle during E13 to E14, which is the main reason for the failure of molting by low expression of *LmCht5-1* and *LmCht5-2*. Finally, we conclude that *LmCht5-1* and *LmCht5-2* were essential for the degradation of the serosal cuticle and the pro-nymphal cuticle. When we control pests in the egg stage, we can reduce the expression of chitinase 5-1 and chitinase 5-2, or reduce the expression of both genes at the same time, to achieve the goal of killing pests during the whole egg stage. All in all, we propose that the effects of *LmCht5-1* and *LmCht5-2* in embryogenesis reveal them as excellent targets for pest control in the egg stage.

## 5. Conclusions

Both *LmCht5-1* and *LmCht5-2* promote the degradation of serosal cuticle and pro-nymphal cuticle during the embryonic development in *L.migratoria*. *dsLmCht5-1* repressed the degradation of chitin between layers both in serosal cuticle and pro-nymphal cuticle. *dsLmCht5-2* inhibited the degradation of chitin to filaments. We propose that the roles of *LmCht5-1* and *LmCht5-2* in embryogenesis indicate that they are excellent targets for pest control during the egg stage.

## Figures and Tables

**Figure 1 biology-11-01778-f001:**
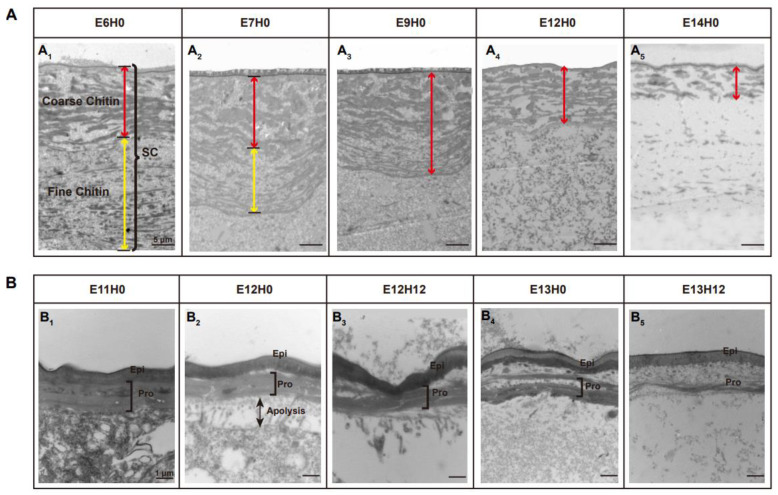
TEM analysis of the ultrastructure of the degradation process of the serosal cuticle and the pro-nymphal cuticle. (**A_1_**): The serosal cuticle is thickest at E6, with the overall stratified structure neatly arranged. The cuticle is composed of a coarse atypical layer on the upper part and a fine atypical layer near the cells on the lower part. (**A_2_**): The serosal cuticle is overall thinner, indicating a degenerated state at E7. The fine parts of the lower layer gradually disappear, while the coarser structures of the upper layer become loose. (**A_3_**): The coarser structure of the serosal cuticle was gradually relaxed at E9. (**A_4_**): The coarser structures on the upper surface continue to degenerate, and the serosal cuticle becomes thinner at E12. (**A_5_**): The coarser structure of the upper layer degenerates into a loose filamentary structure, and the fine chitin structure of the cuticle is almost completely degenerated. Scale bar is 5 μm. SC: serosal cuticle. The red line shows the coarse layer in the upper part, while the yellow line shows the fine chitin layer in the lower part. (**B_1_**): The pro-nymphal cuticle, which consists of an epicuticle and multi-layer procuticle, shows a thickened state at E11. (**B_2_**): The pro-nymphal cuticle begins to degrade at E12. The small gap between the epicuticles and the procuticles indicates the separation of the epicuticles and the procuticles. At the same time, the procuticle separates from the cell, which is called apolysis. (**B_3_**): The epicuticle and procuticle continue to separate at E13, while the layer structure of the procuticle begins to degrade. (**B_4_**): All layer structures of the procuticle are completely loose at E13H12. (**B_5_**): The procuticle of the pro-nymphal cuticle degenerates completely at E14, leaving only the epicuticle. Scale bar is 1 µm. Epi: Epicuticle; Pro: Procuticle.

**Figure 2 biology-11-01778-f002:**
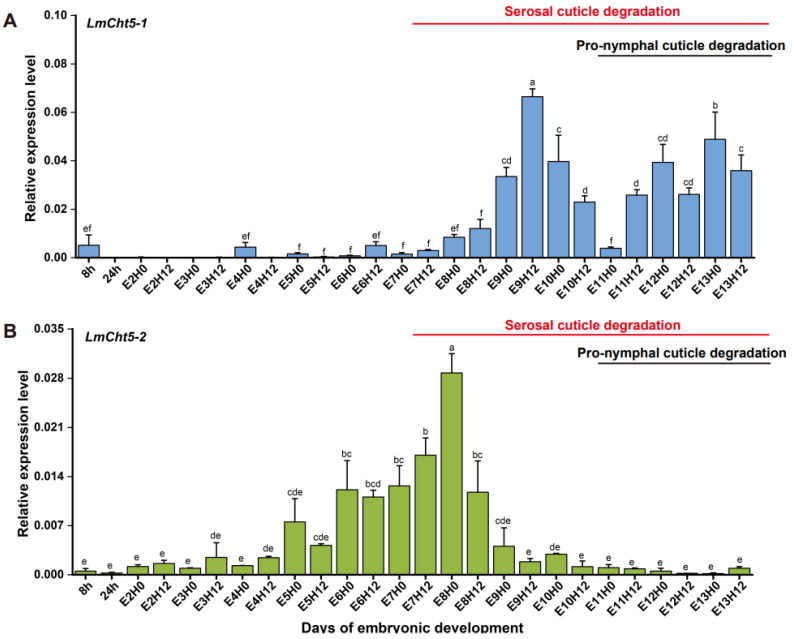
Relative *LmCht5-1* and *LmCht5-2* mRNA transcript levels during locust embryonic development. RT-qPCR analyses of *LmCht5-1* (**A**) and *LmCht5-2* (**B**) transcript levels from (8 h) to (E13H12). Data are represented as means ± SE of three independent biological replications. Different letters represent significant differences between the embryonic developmental. (Tukey’s HSD test, *p* < 0.05). *LmRpl32* was used as the reference gene.

**Figure 3 biology-11-01778-f003:**
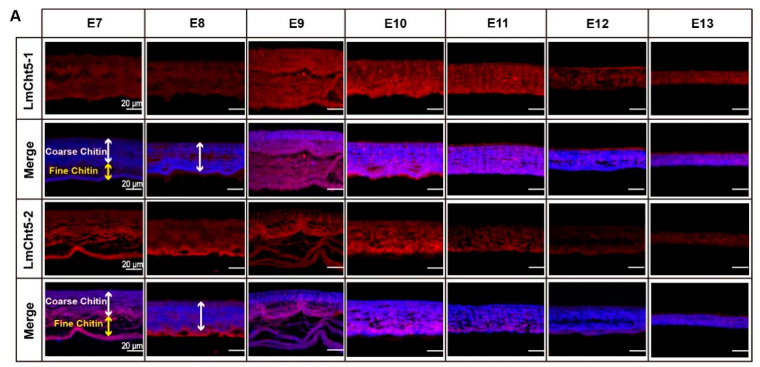
Localization of *LmCht5-1* and *LmCht5-2* in the degradation of the serosal cuticle and pro-nymphal cuticle. Sections (5 µm) of eggs from E7 to E13 were visualized with SYTOX^®^ Green to show the nucleus of cuticle cells (green) and FB28 to counterstain chitin (blue) both in the serosal cuticle (**A**) and the pro-nymphal cuticle (**B**). Localization of *LmCht5-1* and *LmCht5-2* was previously identified by polyclonal antibody detection with the aid of helper Cy3-AffiniPure donkey anti-rabbit antibody (red). *LmCht5-1* is expressed in a highly scattered way in the coarser upper serosal cuticle at E9, while *LmCht5-2* is mainly expressed in the fine structure of the lower serosal cuticle from E7 to E9. During pro-nymphal cuticle degradation, *LmCht5-1* is first located in the cuticle cell at E9 and then expressed in the layer structure of the procuticle during E10 to E13, while *LmCht5-2* kept a lower expression level in the procuticle degradation process. The white line shows the coarse layer in the upper part, while the yellow line shows the fine layer in the lower part. Pnc: Pro-nymphal cuticle; NC: first instar nymphal cuticle.

**Figure 4 biology-11-01778-f004:**
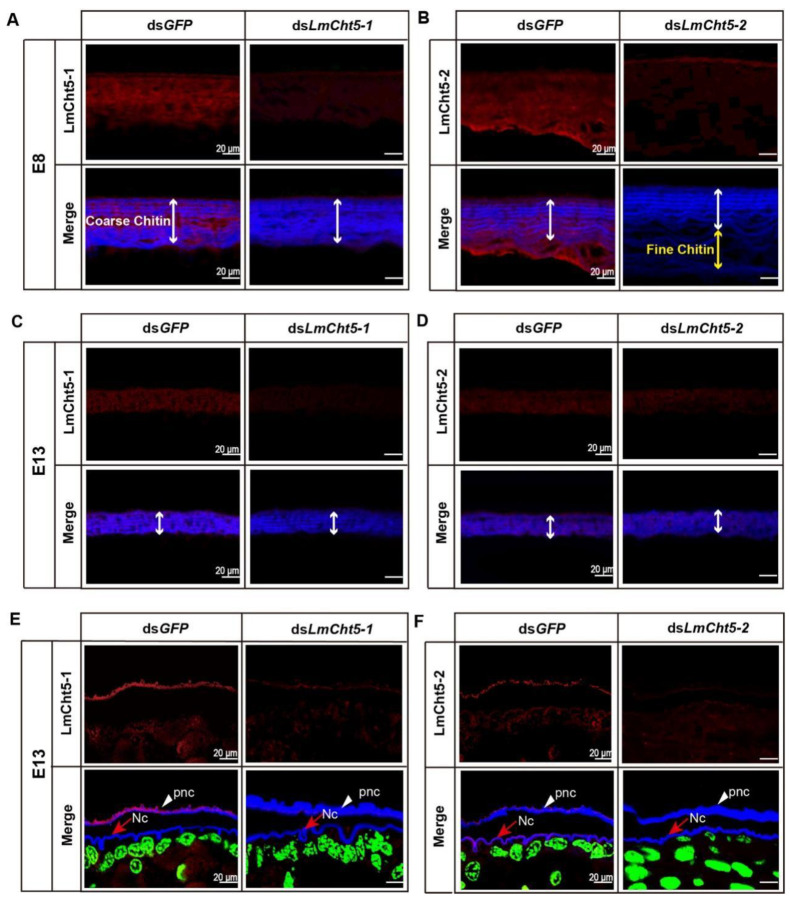
ds*LmCht5-1* and ds*LmCht5-2* inhibit the degradation of chitin both in the serosal cuticle and the pro-nymphal cuticle. Paraffin sections (5 µm) of eggs after RNAi was collected for analysis of chitin and protein differences. (**A**) The degradation of fine structure chitin in the lower part of the serosal cuticle at E8 was not affected by ds*LmCht5-1* microinjection at E5. (**B**) The degradation of fine structure chitin in the lower part of the serosal cuticle at E8 was not obviously inhibited by ds*LmCht5-2* microinjection at E5. (**C**,**D**) The degradation of coarser structure chitin in the upper part of the serosal cuticle at E13 was not obviously affected by ds*LmCht5-1* (**C**) and ds*LmCht5-2* (**D**) microinjection at E7. (**E**) The degradation of chitin in the pro-nymphal cuticle at E13 was significantly inhibited by ds*LmCht5-1* microinjection at E7. (**F**) The degradation of chitin in the pro-nymphal cuticle at E13 was inhibited by ds*LmCht5-2* microinjection to a certain extent at E7. The scale bar is 20 µm. The white line shows the coarse chitin in the upper part, while the yellow line shows the fine layer in the lower part. The blue label indicates chitin, and the green indicates cells. Both *LmCht5-1* and *LmCht5-2* are indicated by the red signal of the antibody. pnc: pro-nymphal; Nc: New cuticle.

**Figure 5 biology-11-01778-f005:**
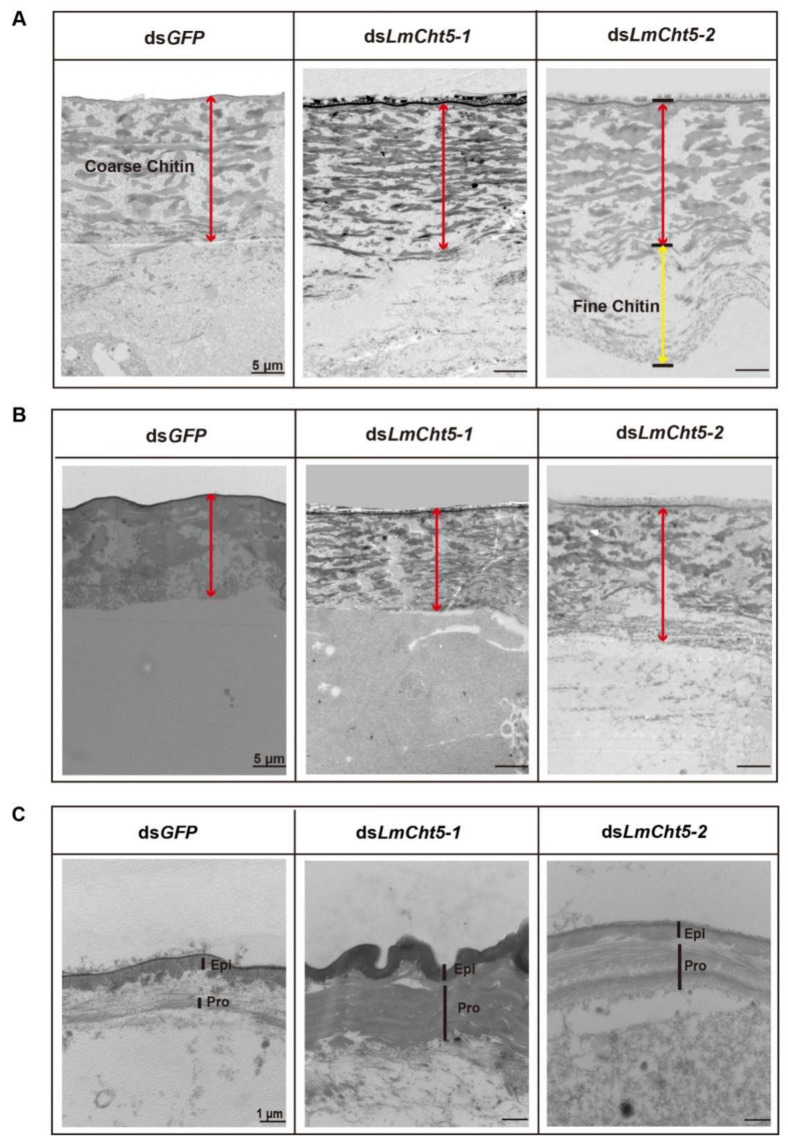
TEM analysis of the function of ds*LmCht5-1* and ds*LmCht5-2* on the degradation of the serosal cuticle and pro-nymph cuticle. (**A**). While ds*LmCht5-1* suppresses the loss of coarser structures on the upper layer of the serosal cuticle, ds*LmCht5-2* inhibits the degradation of fine structures on the lower layer by dsRNA microinjection at E5. (**B**). While ds*LmCht5-1* still suppresses the loss of coarser structures on the upper layer of the serosal cuticle, ds*LmCht5-2* inhibits the further degradation of looser coarser structures at E13 by dsRNA microinjection at E7. (**C**)**.** While ds*LmCht5-1* inhibits the loss of the layer structure of the procuticle, ds*LmCht5-2* suppresses the further degradation of the chitin structure layer at E13 by dsRNA microinjection at E7. Scale bar in (**A**,**B**) is 5 μm. The red line shows the coarse layer in the upper part, while the yellow line shows the fine layer in the lower part. Scale bar in (**C**) is 1 µm. Epi: Epicuticle; Pro: Procuticle.

**Figure 6 biology-11-01778-f006:**
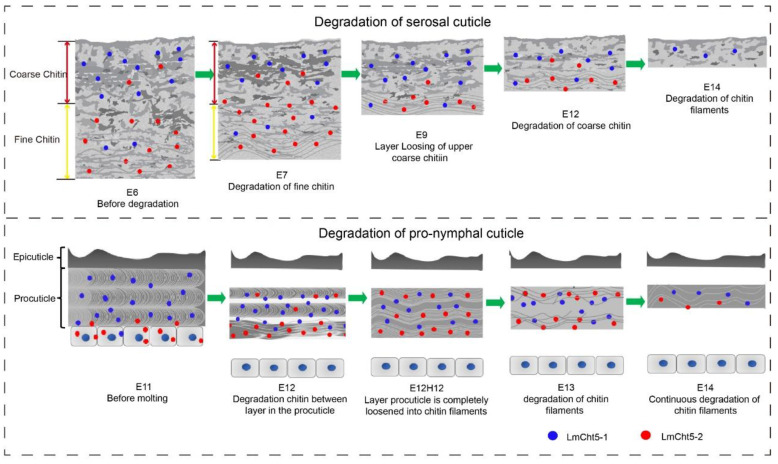
Hypothesis of function of chitinase on chitin degradation in cuticle layers. The degradation of chitin in the cuticle layer is at least two steps. First, the layer connection is loosened by chitinase (*LmCht5-1*) and some unknown hydrolysis X. Then, the chitin inside the layer degrades into filaments mainly by chitinase (*LmCht5-2*) and some unknown hydrolysis Y.

**Table 1 biology-11-01778-t001:** Sequence of primers.

Gene	Application	Primers	Sequence of Primers (5′-3′)
*LmCht5-1*	Gene expression	RT-*LmCht5-1*-F	CATCAAAGCGAAGGGCTACGGC
		RT-*LmCht5-1*-R	AGATTAGTGCGTCCTTCGGGCCA
*LmCht5-2*		RT-*LmCht5-2*-F	ATTTTCAAGGATTATGTGGAGAACC
		RT-*LmCht5-2*-R	TCCACAGTGTTTGTTTTCTTTGATT
*LmRpl32*		RT-*LmRpl32*-F	ACTGGAAGTCTTGATGATGCAG
		RT-*LmRpl32*-R	CTGAGCCCGTTCTACAATAGC
ds*LmCht5-1*	Double-strand	T7-*LmCht5-1*-F	taatacgactcactatagggTCGTTGAGTACATGAAGCGG
	RNA synthesis	T7-*LmCht5-1*-R	taatacgactcactatagggCCTTGTTGATGTAGGTGCCC
ds*LmCht5-2*		T7-*LmCht5-2*-F	taatacgactcactatagggCAGGAAGACTCCTCCACTCG
		T7-*LmCht5-2*-R	taatacgactcactatagggATTCCCAGTCCACGTCAAAG
ds*GFP*		T7-*GFP*-F	taatacgactcactatagggGTGGAGAGGGTGAAGG
		T7-*GFP*-R	taatacgactcactatagggGGGCAGATTGTGTGGAC

Abbreviations: F, forward primer; *GFP*, green fluorescent protein, minuscule showed T7 Promoter; R, reverse primer.

## Data Availability

The data presented in this study are available on request from the corresponding author. The data are not publicly available due to privacy.

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
