# Peer review of "LmCht5-1* and *LmCht5-2* Promote the Degradation of Serosal and Pro-Nymphal Cuticles during Locust Embryonic Development"

_biology, 2022, doi:10.3390/biology11121778_

Round 1
Reviewer 1 Report
The authors have submitted a solid and original work on the chitin degrading enzymes, LmCht5-1 and LmCht5-2, which are expressed during the formation of embryonic and pronymphal cuticles of Locusta embryos. The work follows a logical progression; from EM description of serosal and pro nymphal cuticle morphology, to expression of the two genes, localisation of their products, and characterisation of the phenotype following dsRNAi-mediated knock down. The phenotype seen in cuticles of knockdown embryos is clear and profound. The EM images and the Fluorescence images are all very clear. One of the strengths of the paper is the fine-scale temporal progression shown for the control and knockdown phenotypes. Another strength is the figure summarising the results, hypothesis.
I think the paper could be improved by providing more detail about the motivation for the research. This is limited to a single sentence left at the end of the discussion (“All in all, …”). I’d be interested to know why these genes that regulate embryonic cuticle formation are an important target for pest control, and how this target compares to other pesticide targets.
Minor Points:
1) Should Lagueux have been cited?
AMER. ZOOL., 21:715-726 (1981) Ecdysteroids During Embryogenesis in Locusta migratoria1 MARIE LAGUEUX, CATHERINE SALL, AND JULES A. HOFFMANN
Or Micciarelli?
Journal of Insect Phys. V18:6 1972, Pages 1027-1037
2) There are some parts of the paper that need to be edited for English. Some of them are:
In the first paragraph of the introduction, “times” is used, but should be replaced by “stages”
“Was coated with” should be “secreted”
“When the pronymphal cuticle is shed off, the nymph will be covered by the fourth, nymphal cuticle” should be:
“When the pronymphal cuticle is shed, the nymph will be covered by the fourth embryonic cuticle, or the first nymphal cuticle”
Some suggestions for changes in the Discussion:
“In order to comprehensive understand “ > To obtain a greater understanding
“The more special thing is”> “Specifically”
“In the early embryogenesis” > “During early embryogenesis”
There are a number of examples where the word “loose” or “loosen” are used incorrectly.
Author Response
Answers: We are grateful for your constructive comments on our manuscript. The principle of pest control is that the smaller instar the pest, the better the control effect. So the egg stage is an ideal stage for pest control. Meanwhile, molting is a special phenotype in Arthropoda, including insects, but not in animals. Therefore, controlling the molting process in eggs is a safe and efficient strategy for pest control. So, we added the information on the introduction part.
Meanwhile, we found both LmCht5-1 and LmCht5-2 act at different stages of cuticle degradation and caused embryonic death after interference. Therefore, they can be used as a pest control target during the egg stage, with reducing the expression LmCht5-1 and LmCht5-2 individually, or reducing the expression of two genes at the same time, to achieve the goal of killing pests during the whole egg stage. The related details were added on the discussion part.
Minor Points:
- Should Lagueux have been cited? AMER. ZOOL., 21:715-726 (1981) Ecdysteroids During Embryogenesis in Locusta migratoria1 MARIE LAGUEUX, CATHERINE SALL, AND JULES A. HOFFMANN Or Micciarelli? Journal of Insect Phys. V18:6 1972, Pages 1027-1037.
Answers:
After reading these two references carefully again, we think that from Lagueux M(1972, J.Insect.Phys,18:1027-1037), the main study was the origin and differentiation of the ecdysial glands and the appearance of the embryo cuticles, which providing evidence of a phase relationship between the activity of the ecdysial glands and the embryonic apolysis. However, The paper from Lagueux M (1981, AMER.Zool, 21:715-726) focus on research ecdysteroid molecules, there is not much to describe the involvement of maternal ecdysteroids in the control of crucial events of embryogenesis. Therefore, we cited them from Lagueux M (1979, J.Insect.Phys , 25: 709-723) literature and the 1972 literature together in this paper. Thanks again to the reviewers for their valuable advice.
2) There are some parts of the paper that need to be edited for English. Some of them are: In the first paragraph of the introduction, “times” is used, but should be replaced by “stages”
“Was coated with” should be “secreted”
“When the pronymphal cuticle is shed off, the nymph will be covered by the fourth, nymphal cuticle” should be:
“When the pronymphal cuticle is shed, the nymph will be covered by the fourth embryonic cuticle, or the first nymphal cuticle”
Some suggestions for changes in the Discussion:
“In order to comprehensive understand “ > To obtain a greater understanding
“The more special thing is”> “Specifically”
“In the early embryogenesis” > “During early embryogenesis”
There are a number of examples where the word “loose” or “loosen” are used incorrectly.
Answers:
Thanks for your suggestion. We have carefully revised all of them.

Reviewer 2 Report
Zhang et al. present a study regarding the chitinase genes in L. migratoria. The whole manuscript is very readable and the results are solid as well. I just have two minor issues which should be addressed before an acceptance.
1) I suggest a quantification of the coarse and fine chitin besides the fluorescent images which could be much more comprehensive.
2) Authors need to discuss this work's importance in future pest control scenarios.
Author Response
1) I suggest a quantification of the coarse and fine chitin besides the fluorescent images which could be much more comprehensive.
Answers:
We are grateful to your kindly suggestion for our manuscript. Firstly, both the coarse chitin and fine chitin could just be distinguished by immunofluorescence and transmission electron microscopy. Currently, we do not have the conditions to identify and quantify them in vitro. Secondly, fine chitin only exists for E6-E7 and varies greatly among different samples. Therefore, we marked the different types of chitin with colorful lines to facilitate readers' understanding.
2) Authors need to discuss this work's importance in future pest control scenarios.
Answers:
Thanks for your constructive comments.First of all, we found both LmCht5-1 and LmCht5-2 could degrade the cuticle at different stages and caused embryonic death after interference. Therefore, they can be used as a pest control target during the egg stage, with reducing the expression LmCht5-1 and LmCht5-2 individually, or reducing the expression of two genes at the same time, to achieve the goal of killing pests during the whole egg stage. The related details were added on the discussion part.
